

# Evaluation of potential reference genes for quantitative RT-PCR analysis in spotted sea bass (*Lateolabrax maculatus*) under normal and salinity stress conditions

Haolong Wang[1,2], Haishen Wen[1,2], Yun Li[1,2], Kaiqiang Zhang[1,2] and Yang Liu[1,2]

[1] College of Fisheries, Ocean University of China, Qingdao, China
[2] The Key Laboratory of Mariculture (Ocean University of China), Ministry of Education, Ocean University of China, Qingdao, China

## ABSTRACT

The aim of this study was to select the most suitable reference genes for quantitative real-time polymerase chain reaction (qRT-PCR) of spotted sea bass (*Lateolabrax maculatus*), an important commercial marine fish in Pacific Asia, under normal physiological and salinity stress conditions. A total of 9 candidate reference genes (*HPRT*, *GAPDH*, *EF1A*, *TUBA*, *RPL7*, *RNAPol II*, *B2M*, *ACTB* and *18S rRNA*) were analyzed by qRT-PCR in 10 tissues (intestine, muscle, stomach, brain, heart, liver, gill, kidney, pectoral fins and spleen) of *L. maculatus*. Four algorithms, geNorm, NormFinder, BestKeeper, and comparative ΔCt method, were used to evaluate the expression stability of the candidate reference genes. The results showed the *18S rRNA* was most stable in different tissues under normal conditions. During salinity stress, *RPL7* was the most stable gene according to overall ranking and the best combination of reference genes was *RPL7* and *RNAPol II*. In contrast, *GAPDH* was the least stable gene which was not suitable as reference genes. The study showed that different algorithms might generate inconsistent results. Therefore, the combination of several reference genes should be selected to accurately calibrate system errors. The present study was the first to select reference genes of *L. maculatus* by qRT-PCR and provides a useful basis for selecting the appropriate reference gene in *L. maculatus*. The present study also has important implications for gene expression and functional genomics research in this species or other teleost species.

## INTRODUCTION

Quantifying gene expression levels is an essential research strategy to understand and reveal complex regulatory gene networks in organisms (*Dekkers et al., 2012*). Quantitative real-time PCR (qRT-PCR) is considered the most powerful and commonly used tool for analyzing the relative transcription levels in gene expression because of its advantages of easy accessibility, high-throughput and fast-processing. Additionally, qRT-PCR can

Corresponding author
Yun Li, yunli0116@ouc.edu.cn

detect the low abundance of transcripts and small changes in gene expression. To obtain reliable gene expression profiles, accurate transcript normalization by using the internal reference genes (housekeeping genes) is a necessary prerequisite. The optimal reference genes should be constant with the adjustment of the experimental procedure (tissues, treatments and developmental stages) (*Radonić et al., 2004*). However, the stability of the reference gene is relative, and the expression level of the reference gene used might commonly be unstable under different conditions (*Gutierrez et al., 2008*). No single gene maintained constant expression levels in all species and different tissues and under different experimental conditions. For example, *β-Actin* and *UBCE* were the most stable genes in Japanese flounder (*Paralichthys olivaceus*) tissues, whereas *18S rRNA* showed the most stable expression in all embryonic developmental stages in *P. olivaceus* (*Zheng & Sun, 2011*; *Zhong et al., 2008*). Therefore, it is necessary to select specific reference genes of species and tissues that could be helpful for studies on regulatory gene networks under different conditions.

The spotted sea bass, *Lateolabrax maculatus*, is a newly redescribed species (*Yokogawa & Seki, 1995*; *Kim et al., 2001*; *Liu et al., 2006*; *Yokogawa, 2013*) with commercially significant value in the recreational fishery and mariculture industry in Pacific Asia. This fish is widely distributed along the Chinese coast, reaching south to the borders of Vietnam and north to Korea (*Yokogawa & Seki, 1995*). Furthermore, as euryhaline species, the spotted sea bass is a valued fish model that could be cultured in fresh water after domestication. To date, considering its economic value, the expression levels of several potential trait-related function genes have been reported in *L. maculatus*, such as hypothalamus-pituitary-gonad (HPG) axis genes (*Wang et al., 2017*), heat shock protein 70 (HSP70) genes (*Han et al., 2017*). Moreover, the salinity stress-responsive transcriptome has also been analyzed (*Zhang et al., 2017*). In these reports, *β-actin* and *18S rRNA*, as traditional reference genes, have been used without validation for appropriateness. Moreover, *β-actin*, as an internal standard for gene expression quantitation, could show confounding results (*Glare et al., 2002*). Thus far, there is no validated reference gene reported in *L. maculatus*. Thus, it is necessary to identify and select suitable reference genes for the accurate analysis of gene expression in *L. maculatus*.

Stress in fish caused by abiotic factors encountered in nature and aquaculture, such as acid–base, salinity and temperature, leads to various responses that might be adaptive or maladaptive. Among these responses, salinity is a major abiotic factor that affects the growth, hatch, reproduction and survival of fish species (*Imsland et al., 2001*; *Tandler, Anav & Choshniak, 1995*; *Berlinsky et al., 2004*). The ability to endure changes in salinity depends on the capacity to regulate osmotic pressure (*Tandler, Anav & Choshniak, 1995*). Moreover, the gill, kidney and intestine are important osmoregulatory organs in fish to maintain the balance of ionic composition and osmolality of the fluid in teleosts (*Katoh et al., 2000*). Particularly, the gill is a functionally and morphologically complex tissue comprising plentiful, interconnected physiological activities, which are vital to maintaining systemic homeostasis in the face of changing internal and external environments (*Evans, Piermarini & Choe, 2005*). For this reason, the present study was aimed to select suitable reference genes and evaluate the reference genes stability in *L. maculatus* among different tissues and

under the salinity stress. A total of 9 reference genes, including *18s rRNA*, *HPRT*, *GAPDH*, *EF1A*, *TUBA*, *RPL7*, *RNAPol II*, *B2M* and *ACTB*, were selected. The present study could provide some theoretical basis for selecting reference genes in *L. maculatus* and other fishes.

## MATERIAL AND METHODS

### Animals, treatments and fish sampling

All animal experiments were conducted in accordance with the guidelines and approval of Institutional Animal Care and Use Committee of Ocean University of China. The field studies did not involve endangered or protected species.

Spotted sea bass (786.53 ± 18.28 g), cultured in cages, were obtained from Jiaonan (Qingdao City, Shandong Province, China) and then transported to the laboratory. The fish were acclimatized at room temperature in seawater (30 ppt) with continuous aeration for a week prior to the experiment. Nine healthy fish were randomly divided into 3 groups as three biological replicates. The fish were treated with tricaine methanesulfonate (MS 222, 200 mg/L) and immediately dissected. The intestine, muscle, stomach, brain, heart, liver, gill, kidney, pectoral fins and spleen were collected. Ten tissues per fish were numbered and stored at −80 °C for RNA extraction.

For the salinity challenge experiment, 60 spotted sea bass (100.00 ± 2.34 g) were acquired from Shuangying Aquatic Seed Company (Lijin County, Dongying City, Shandong Province). The fish were acclimatized at a density of 5 individuals per tank (120 L). Water temperature, pH, dissolved oxygen and light-dark cycle were maintained at 21°C (±0.5 °C), 7.98∼8.04, 6.90∼8.54 mg/L and 14 L:10 D, respectively. After acclimation, the fish were randomly divided into 4 groups with different salinities (0, 12, 30, and 45 ppt). After rearing for 30 days, 9 fishes per group were randomly selected as three biological replicates and treated with MS 222 (200 mg/L). Gill tissues were immediately frozen in liquid nitrogen and then stored at −80 °C until further use.

### RNA extraction and cDNA synthesis

Two experimental sample sets were constructed. Set A: different tissues (intestine, muscle, stomach, brain, heart, liver, gill, kidney, pectoral fins, and spleen) and set B: different salinities (0, 12, 30, and 45 ppt). The total RNA was extracted from samples by using TRIzol reagent (Invitrogen, USA) according to the manufacturer's instructions and digested with RNase-free DNase I (TaKaRa, Japan) to remove genomic DNA contamination. Equal amounts of RNA from the same tissues of 3 individual fish under the same conditions were pooled as one sample to minimize the variation among individuals, and three such pools were obtained for each tissue and salinity treatment group. The concentration was determined by nucleic acid protein analyzer BD1000 (Beijing, China), and the quality of RNA was assessed by gel electrophoresis. Samples with 260/280 absorbance ratios greater than 1.9 were used for cDNA synthesis. A 0.5-μg aliquot of total RNA from each sample was reverse transcribed by using the PrimeScript^TM RT reagent Kit with gDNA Eraser (TaKaRa, Japan) employing a RT Primer MIX (Random 6 mers and Oligo dT Primer) in a 20 μl reaction according to the manufacturer's instructions. The synthesized cDNA was stored at −20 °C.
**Table 1  Summary of reference genes in this study.**

| Abbreviation | Reference gene name | NCBI accession number |
|---|---|---|
| HPRT | Hypoxanthine guanine phosphoribosyl transferase1 | MH181802 |
| GAPDH | Glyceraldehyde-3-phosphate dehydrogenase | MH181799 |
| EF1A | Elongation factor-1-$\alpha$ | MH181801 |
| TUBA | $\alpha$-Tubulin | MH181800 |
| RPL7 | Ribosomal protein L7 | MH181805 |
| RNAPol II | RNA polymerase II subunit C | MH181803 |
| B2M | $\beta$-2-microglobulin | MH181798 |
| ACTB | $\beta$-Actin | MH181804 |
| 18S rRNA | 18S ribosomal RNA | JN211898 |

## Selection of reference genes for spotted sea bass

Nine reference genes were selected for gene expression analysis, including the reference gene (*18S rRNA*) sequences from the GenBank database (https://www.ncbi.nlm.nih.gov/genbank/) and 8 reference gene sequences from the transcriptomics database by the IlluminaHiseq 4,000 platform (*Zhang et al., 2017*). The abbreviated and full gene names and the GenBank accession numbers are provided in Table 1.

## Primers design and qRT-PCR

All reference gene primers were designed by the Primer 5.0 software. Nine primer pairs were synthesized by the Beijing Genomics Institute (BGI) and tested via standard RT-PCR by using six serial five-fold dilutions of sample cDNA with SYBR® Premix Ex Taq$^{TM}$ (TaKaRa, Japan). The specificity of amplification was verified by melting curve and agarose gel electrophoresis, and the primer amplification efficiency was calculated as $E(\%) = (10^{(-1/\text{slopes})} - 1) \times 100$ (*Pfaffl, 2001*). The qRT-PCR was performed in 96-well plates by using the StepOne Plus Real-Time PCR system (Applied Biosystems). The reaction (20 µl) was performed by using SYBR® Premix Ex Taq$^{TM}$ (TaKaRa, Japan). Each well contained 10 µl of SYBR® Premix Ex Taq$^{TM}$, 0.4 µl of ROX Reference Dye, 6.8 µl of sterilized ddH2O, 0.4 µl of each primer (10 µmol L$^{-1}$), and 2 µl of cDNA template. The reaction conditions were 95 °C for 30 s, followed by 40 cycles at 95 °C for 5 s and 60 °C for 30 s. After PCR amplification, a melting curve was obtained by the following process: 95 °C for 5 s, 60 °C for 1 min, followed by 95 °C at the rate of 0.11 °C per second to verify primer specificity. All RT-qPCR assays were carried out in three biological replicates with three technical replicates.

## Statistical analysis

The expression stability of the 9 reference genes was evaluated by 4 different algorithms: geNorm (*Vandesompele et al., 2002*), NormFinder (*Andersen, Jensen & Orntoft, 2004*), BestKeeper (*Pfaffl et al., 2004*), and comparative ΔCt method (*Silver et al., 2006*). The comprehensive ranking of candidate reference genes was evaluated by calculating the geometric mean of each reference gene ranking (*Chen et al., 2011*). The raw Ct values in geNorm and NormFinder were previously transformed to relative quantities (RQ).

**Table 2  Primer sequences, product sizes and PCR efficiencies of the selected genes.**

| Gene name | 5′–3′ primer sequence | Amplicon size (bp) | Primer efficiency (%) | Correlation coefficients |
|---|---|---|---|---|
| HPRT-F | TGCTCAAAGGGGGTTACAAG | 117 | 105.74 | 0.9966 |
| HPRT-R | AGTAGCTCTTGAGGCGGATG | | | |
| GAPDH-F | AGCTCAATGGCAAGCTGACT | 125 | 94.16 | 0.9994 |
| GAPDH-R | GGCCTTCACAACCTTCTTGA | | | |
| EF1A-F | GCAAGTTCAGGGAGCTCATC | 121 | 99.44 | 0.9976 |
| EF1A-R | ATTGGCTTCTGTGGAACCAG | | | |
| TUBA-F | AGGTCTCCACAGCAGTAGTAGAGC | 89 | 106.67 | 0.9993 |
| TUBA-R | GTCCACCATGAAGGCACAGTCG | | | |
| RPL7-F | ACCCCAACCTGAAGTCTGTG | 121 | 101.11 | 0.9986 |
| RPL7-R | ATGCCATATTTGCCAAGAGC | | | |
| RNAPol II-F | GTCAGGAACTACGGCTCAGG | 117 | 102.88 | 0.9975 |
| RNAPol II-R | TGTGCCTCAGTGCATTGTCT | | | |
| B2M-F | GACCTGGCCTTCAAACAGAA | 125 | 102.05 | 0.9993 |
| B2M-R | TCCCAGGCGTAATCTTTGAC | | | |
| ACTB-F | CAACTGGGATGACATGGAGAAG | 114 | 99.46 | 0.9981 |
| ACTB-R | TTGGCTTTGGGGGTTCAGG | | | |
| 18S rRNA-F | GGGTCCGAAGCGTTTACT | 179 | 94.31 | 0.9969 |
| 18S rRNA-R | TCACCTCTAGCGGCACAA | | | |

$RQ = (1 + E)^{\Delta Ct}$, $\Delta Ct = $ lowest Ct value—Ct value of sample. E is equal to 2 when PCR efficiencies approach 100%. The highest relative quantities for each gene are set to 1. Finally, all the reference genes were ranked by four programs. Then, all graphs were generated by using SPSS 19.0 and OriginPro 8.0.

## RESULTS

### Amplification efficiencies of primers

A single peak was obtained in each amplification during the analysis of the melting curves after 40 cycles by the Applied Biosystems StepOne Plus Real-Time PCR system (Fig. S1), and agarose gel electrophoresis showed that each of the amplifications products was a single band of the expected size (Fig. S2). The primer efficiency (E) and correlation coefficients ($R^2$) were determined based on the standard curves. The primer efficiency (E) of the nine genes ranged from 94.16% to 106.67%, and the correlation coefficients ($R^2$) ranged from 0.9966 to 0.9994 (Table 2).

### Transcription levels of candidate reference genes

The transcription levels of all 9 candidate reference genes were assessed by qRT-PCR. The raw Ct values showed different variation and transcription levels. In different tissues, the coefficient of variation (CV) of the raw Ct values was calculated to evaluate transcription level variations. The CV of all reference genes ranged from 4.011 to 17.550%, and the Ct values varied from 9.402 to 34.015. *GAPDH* and *RNAPol II* were the most variable and the least variable reference genes, respectively. Among these reference genes, *18S rRNA* showed
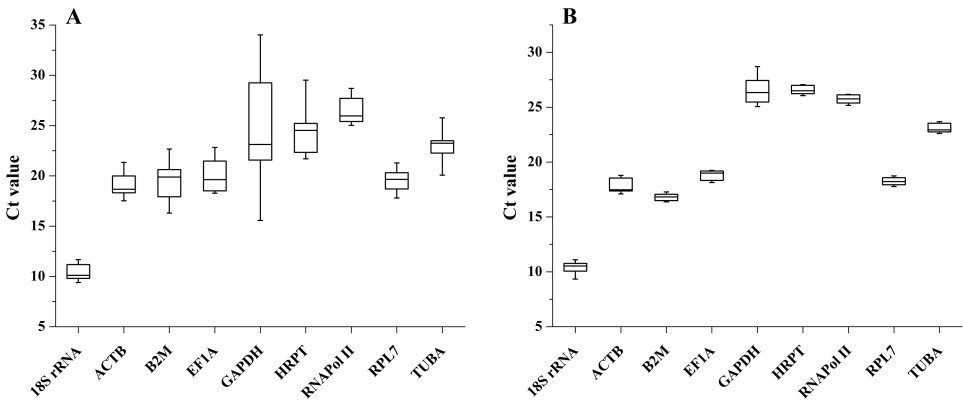

**Figure 1 Expression levels of candidate reference genes in different tissues (A) and salinity stress (B).** The boxes indicate the 1st and 3rd quartiles. The vertical lines (whiskers) represent the maximum and minimum values.

the highest transcription level (average Ct = 10.360), while *RNAPol II* showed the lowest expression level (average Ct = 26.399). The expression trend of these candidate reference genes in the samples after salinity treatment was similar to that in different tissues, and the CV of all reference genes ranged between 1.078 to 3.830% and Ct values varied from 9.661 to 28.133. *GAPDH* displayed the highest transcription-level variation. The lowest transcription level variation was observed for *HRPT*, followed by *RPL7* and *RNAPol II*. The minimum mean Ct value was 10.386, and the maximum mean Ct value was 26.517 for the highest and lowest expression levels for *18SrRNA* and *GAPDH* (Fig. 1).

## Evaluation of stability of the candidate reference genes

To select optimal reference genes for accurate normalization under the same experimental conditions, four common algorithms were used to analyze expression stability and rank the reference genes.

### *geNorm analysis*

geNorm defined the M value as the expression stability measure, which describes the average pairwise variation of a candidate gene relative to all other candidate genes. The tested sample gene with lowest M value shows the most stable expression and vice versa. In the present study, the expression stability M value of 9 candidate reference genes was calculated by the geNorm program. Among different tissues, *18S rRNA* and *ACTB* have the least M value of 0.90, while *GAPDH* showed the highest value, indicating that *18S rRNA* and *ACTB* were most stable in expression and that *GAPDH* was the least stable in expression. Furthermore, on the basis of geNorm analysis, the default limit of the stability value (M) is <1.5; thus, *GAPDH*, *HRPT* and *B2M*, with stability values (M) above 1.5, were not selected in gene expression. However, under salinity stress, the stability values (M) of all candidate reference genes were below 1.5. *RNAPol II* and *TUBA* were the most stable genes, with M values of 0.16, while *GAPDH* was the least stable gene, with an M value of 0.67. Thus, the geNorm analysis indicated that *18S rRNA* and *ACTB* were the most stable

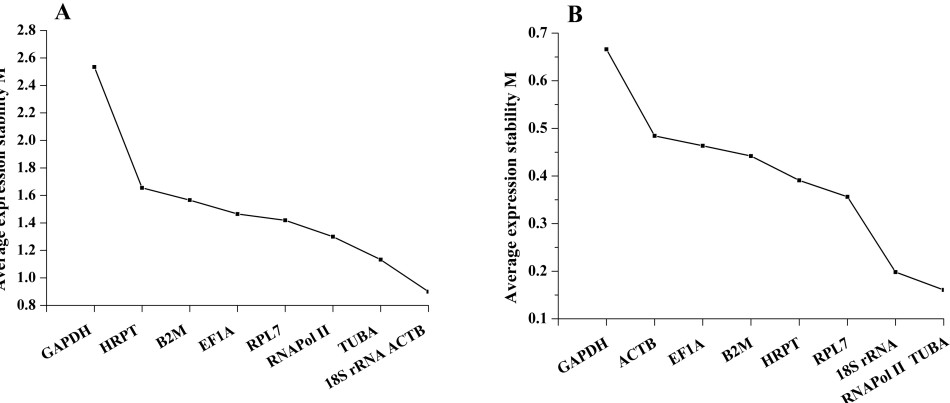

**Figure 2** Average expression stability values of the candidate reference genes (A) in different tissues and (B) under salinity stress analyzed by geNorm.

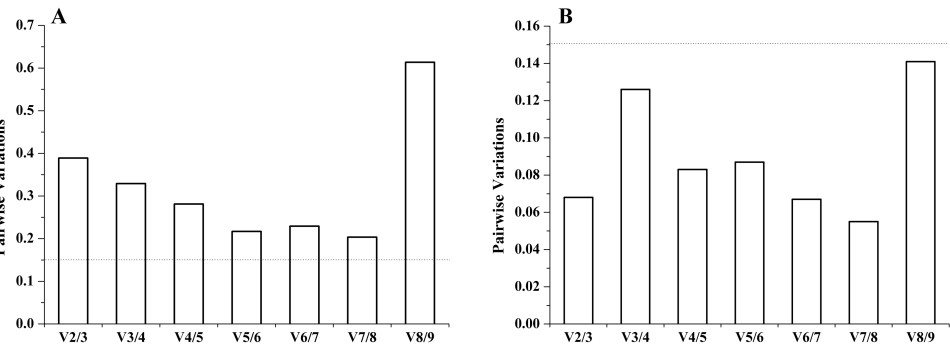

**Figure 3** The number of reference genes calculated by geNorm in different tissues (A) and under salinity stress (B). The dotted lines represent the cut-off limit value of 0.15.

reference genes among different tissues, and *RNAPol II* and *TUBA* were the most stable genes in samples under salinity treatments, whereas *GAPDH* was the least stable gene in both experimental sets (Fig. 2).

It is important to determine the optimal number of genes for accurate normalization in qRT-PCR. The geNorm algorithm was based on the analysis of the pairwise variation ($V_n/V_{n+1}$) of sequential normalization factors to determine the optimal number of reference genes. The cutoff limit was set as 0.15 for pairwise variation, below which the addition of more genes is not necessary. For both experimental sets, in different tissues, all the pairwise variation ($V_n/V_{n+1}$) was above 0.15. The inclusion of the sixth gene had approximately the same effect ($V_5/V_6 = 0.217$) on the NF as the inclusion of the eighth gene ($V_7/V_8 = 0.204$) had. Therefore, no stable combination was desirable for the selected reference genes in different tissues. However, under salinity stress, all pairwise variation (Vn/Vn+1) was below 0.15, and the $V_2/V_3$ was well below 0.15. Thus, two genes (*RNAPol II* and *TUBA*) had the optimal number combination under salinity stress (Fig. 3).

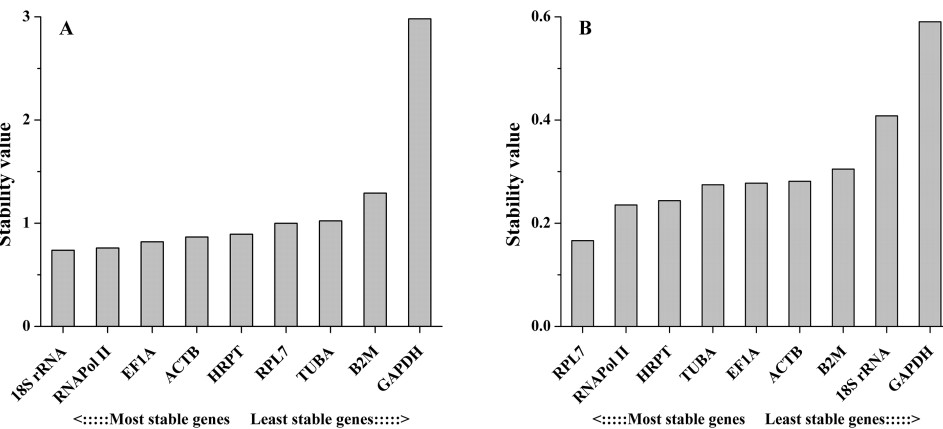

**Figure 4** Average expression stability values of the candidate reference genes in different tissues (A) and under salinity stress (B) analyzed by NormFinder.

### NormFinder analysis

The NormFinder algorithm could also estimate the expression stability and rank the genes according to stability, which was based on the estimation of intra- and inter-group variations. The gene with the lowest stability value is considered to show the most stable expression and vice versa. As the NormFinder analysis showed, the optimal combinations were similar to the results obtained by geNorm, with slight differences. In different tissues, *18S rRNA* (0.738) was the most stable gene, followed by *RNAPol II* (0.760), while *GAPDH* (2.981) was the least stable gene (Fig. 4A). During salinity stress, *RPL7* (0.166) and *RNAPol II* (0.235) were the most stable genes, while *GAPDH* (0.590) was the least stable gene (Fig. 4B). The estimation of intra- and inter-group variations was also obtained (Table S1).

### BestKeeper analysis

The BestKeeper algorithm estimates the expression stability of candidate genes by calculating and comparing the variation, including the coefficient of variance (CV) and standard deviation (SD). The most stable reference gene was selected based on the size of the SD value. The SD values of the candidate reference genes were negatively correlated with the stability of the gene, indicating that the lowest SD value shows the highest stability. Similarly, the results for different tissues by BestKeeper were highly similar to those obtained by geNorm and NormFinder. For example, *18S rRNA* was the most stable gene, followed by *ACTB*, with an SD value <1, while *GAPDH*, with an SD value of 4.357, had the lowest stability. However, *RPL7* was identified as the most stable gene, whereas *GAPDH* was the least stable gene under salinity stress (Table 3).

### Comparative ΔCt method

The comparative ΔCt method identifies optimal candidate genes by comparing the relative expression of a pair of genes in each sample. If the ΔCt value between the two genes remains constant, then both genes are stable. However, if the ΔCt value fluctuates, then

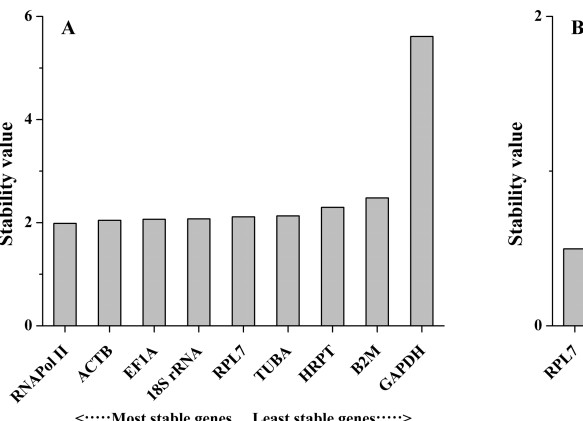
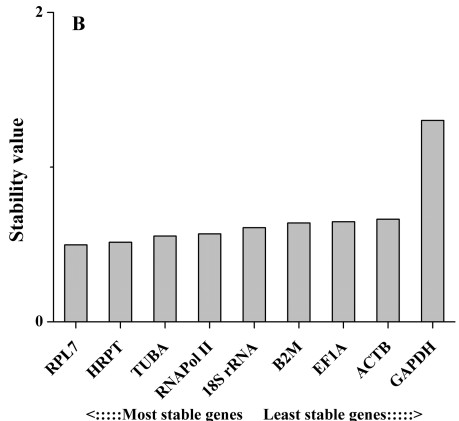

**Figure 5** Stability values of the candidate reference genes in different tissues (A) and under salinity stress (B) analyzed by Comparative ΔCt method.

**Table 3** Descriptive statistics of 9 candidate reference genes based on their quantification cycle values analyzed by BestKeeper.

|  | Parameters | Genes | | | | | | | | |
|---|---|---|---|---|---|---|---|---|---|---|
|  |  | *18SrRNA* | *ACTB* | *B2M* | *EF1A* | *GAPDH* | *HRPT* | *RNAPolII* | *RPL7* | *TUBA* |
| Different tissues *n* = 90 | Geo mean [CP] | 10.335 | 19.008 | 19.451 | 19.979 | 24.277 | 24.392 | 26.372 | 19.514 | 22.896 |
|  | Ar mean [CP] | 10.360 | 19.038 | 19.541 | 20.038 | 24.828 | 24.481 | 26.399 | 19.542 | 22.941 |
|  | Min [CP] | 9.402 | 17.527 | 16.305 | 18.275 | 15.573 | 21.691 | 25.023 | 17.809 | 20.080 |
|  | Max [CP] | 11.670 | 21.344 | 22.671 | 22.831 | 34.015 | 29.517 | 28.703 | 21.293 | 25.779 |
|  | Std dev [± CP] | 0.628 | 0.877 | 1.453 | 1.366 | 4.357 | 1.498 | 1.059 | 0.898 | 1.035 |
|  | CV [% CP] | 6.061 | 4.604 | 7.436 | 6.817 | 17.550 | 6.121 | 4.011 | 4.594 | 4.511 |
| Different salinities *n* = 36 | Geo mean [CP] | 10.376 | 17.812 | 16.790 | 18.817 | 26.494 | 26.480 | 25.769 | 18.095 | 23.136 |
|  | Ar mean [CP] | 10.386 | 17.821 | 16.793 | 18.821 | 26.517 | 26.482 | 25.772 | 18.096 | 23.140 |
|  | Min [CP] | 9.661 | 17.262 | 16.421 | 18.163 | 25.222 | 26.145 | 25.218 | 17.856 | 22.584 |
|  | Max [CP] | 10.828 | 18.783 | 17.204 | 19.208 | 28.133 | 26.936 | 26.106 | 18.418 | 23.631 |
|  | Std dev [± CP] | 0.363 | 0.481 | 0.257 | 0.329 | 1.016 | 0.285 | 0.306 | 0.202 | 0.396 |
|  | CV [% CP] | 3.491 | 2.697 | 1.532 | 1.749 | 3.830 | 1.078 | 1.188 | 1.114 | 1.710 |

one or both genes have unstable expression. In different tissues, the results obtained by this method were similar to those obtained with the other methods, with a few exceptions. For example, *18S rRNA* was ranked the 4th most stable reference gene by this method (Fig. 5A). However, this gene was ranked the most stable reference gene by geNorm, NormFinder and BestKeeper methods. Under salinity stress, *RPL7* showed the least variation, followed by *HRPT*. The least stable gene in both experimental sets was *GAPDH* (Fig. 5B).

### Recommended comprehensive ranking

Based on the rankings results from four algorithms, the overall ranking of reference genes was obtained. The geometric mean of each reference gene ranking was calculated for the overall final ranking. For example, *RPL7* ranked the 4th, 6th, 3th, and 5th place among different tissues in geNorm, NormFinder, BestKeeper, and comparative ΔCt method,

**Table 4  Ranking of candidate reference genes by geNorm, NormFinder, BestKeeper, comparative ΔCt method, and overall rank.**

| Conditions | Ranking | geNorm rank | NormFinder rank | BestKeeper rank | ΔCt rank | overall rank |
|---|---|---|---|---|---|---|
| Tissue | 1 | 18S rRNA / ACTB | 18S rRNA | 18S rRNA | RNAPol II | 18S rRNA |
| | 2 | TUBA | RNAPol II | ACTB | ACTB | ACTB |
| | 3 | RNAPol II | EF1A | RPL7 | EF1A | RNAPol II |
| | 4 | RPL7 | ACTB | TUBA | 18S rRNA | EF1A |
| | 5 | EF1A | HRPT | RNAPol II | RPL7 | TUBA |
| | 6 | B2M | RPL7 | EF1A | TUBA | RPL7 |
| | 7 | HRPT | TUBA | HRPT | HRPT | HRPT |
| | 8 | GAPDH | B2M | B2M | B2M | B2M |
| | 9 | | GAPDH | GAPDH | GAPDH | GAPDH |
| Salinity stress | 1 | RNAPol II / TUBA | RPL7 | RPL7 | RPL7 | RPL7 |
| | 2 | 18S rRNA | RNAPol II | B2M | HRPT | RNAPol II |
| | 3 | RPL7 | HRPT | HRPT | TUBA | HRPT |
| | 4 | HRPT | TUBA | RNAPol II | RNAPol II | TUBA |
| | 5 | B2M | EF1A | EF1A | 18S rRNA | B2M |
| | 6 | EF1A | ACTB | 18S rRNA | B2M | 18S rRNA |
| | 7 | ACTB | B2M | TUBA | EF1A | EF1A |
| | 8 | GAPDH | 18S rRNA | ACTB | ACTB | ACTB |
| | 9 | | GAPDH | GAPDH | GAPDH | GAPDH |

respectively. Then, the geometric mean of the four ranking numbers was calculated, thus for, *RPL7* the geometric mean is 4.36 $[(4 \times 6 \times 3 \times 5)^{0.25}]$. The gene with the lowest geometric mean shows the highest stability. As shown in Table 4, in different tissues, *18S rRNA > ACTB > RNAPol II > EF1A > TUBA > RPL7 > HRPT > B2M > GAPDH*. Under salinity stress, *RPL7 > RNAPol II > HRPT > TUBA > B2M > 18S rRNA > EF1A > ACTB > GAPDH*.

## DISCUSSION

The qRT-PCR is a highly sensitive, specific and reproducible method for gene expression analysis. The optimal reference gene is constantly transcribed in different types of cells, tissues, and species and under various experimental conditions. However, the most stable reference gene to meet all conditions is almost non-existent. The selection of a proper reference gene is the precondition for the accurate analysis of the expression level of a target gene in quantitative real-time PCR. Thus far, the expression levels of the currently used reference genes showed large differences under various treatment conditions. For example, the expression level of *GAPDH* showed a significant difference in black rockfish(*Sebastes schlegeli*) during larvae developmental stages and tissue analysis (*Ma et al., 2013*). A number of common reference genes have been used without being validated. Therefore, to avoid unnecessary errors in the profiling of gene expression, the expression stability of 9 candidate reference genes in different tissues and under salinity stress was analyzed by four programs (geNorm, NormFinder, BestKeeper and comparative

ΔCt method). The four types of algorithms showed that *GAPDH* was least stable gene in common. However, there are some differences in the ranking order of stability. For example, among different tissues, *RPL7* was ranked the 4th most stable reference gene by geNorm and 5th by comparative ΔCt method but ranked 6th by NormFinder and 3rd by BestKeeper. In general, the differences in these results might be due to the different algorithms among these applications. Similar results have also been observed in several studies (*Bower & Johnston, 2009*; *Urbatzka et al., 2013*). However, there is no consensus on which application is better to use.

In the present study, *18S rRNA* was the most suitable gene in different tissues when using qRT-PCR for RNA transcription analysis. Similarly, *18S rRNA* was one of the most stable genes in seven tissues of Nile tilapia (*Yang et al., 2013*), and *EF1 α*, *Rpl13 α* and *18S rRNA* were more suitable as a reference gene panel for zebrafish tissue analysis (*Tang et al., 2007*). Moreover, *18S rRNA* was a classical reference gene and has been described as a preferable control (*Blanquicett et al., 2002*). Conversely, the study (*Radonić et al., 2004*; *Fernandes et al., 2008*) showed that *18S rRNA* was not suitable for internal reference genes. *18S rRNA* transcription could display changes in gene expression related to the imbalance between messenger and ribosomal RNA content in rat mammary tumors (*Solanas, Moral & Escrich, 2001*). In addition, *18S rRNA* had a markedly high transcription level compared to that of other genes, which indicates that cDNA samples need larger template dilutions within the dynamic range of qRT-PCR, particularly when the target gene expression level is weak. The transcript abundance of the reference gene may affect the results of gene expression (*Filby & Tyler, 2007*). Nevertheless, the use of *18S rRNA* is highly recommended as an internal control standard in tissues for target gene expression, and *ACTB* may be an appropriate choice when the target gene is not abundant in expression level. Interestingly, in Asian seabass (*Lates calcarifer*) (*Paria et al., 2016*), *ACTB* and *EF1A* are the most stable genes across the tissues of normal animals and *18S rRNA* and *EF1A* are the best reference genes in bacteria challenged animals. This is roughly the same as the results of our present study. The slight difference may be due to differences in experimental conditions and the number of algorithm programs.

Salinity is one of the most important environmental factors for aquatic organisms. The transcription abundance was measured from low to high salinity. As a member of the ribosomal protein family, *RPL7* was considered a suitable gene in salinity stress in the present study. This gene was also confirmed in other studies. For example, *RPL7* was the most stable gene in the liver of zebrafish under bacterial expression. *Varsamos et al. (2006)* reported similar findings in European seabass (*Dicentrarchus labrax*), showing that *RPL17* was a valid candidate references in seawater and following acclimation to fresh water. Similarly, *L13a* (*RPL13a*) are recommended for qPCR normalization according to BestKeeper and NormFinder (*Mitter et al., 2009*). For an optimal number of reference genes, geNorm analysis suggests the inclusion of one or more genes for accurate normalization when the cut off range of the pairwise variation value is above 0.15. In the present study, the pairwise variation was above 0.15 in different tissues. However, 0.15 is not an absolute cutoff value but rather an ideal value depending on the number of genes and types of samples tested (*Singh et al., 2015*). We agree that more than one gene should be

used as a reference gene for calibration in the normalization process. Therefore, more genes should be selected as candidate reference genes when studying the gene expression related to various tissues. In addition, with in-depth whole genome sequencing, the acquisition of reference gene will no longer be limited to a few traditional reference genes.

## CONCLUSION

In the present study, we evaluated the stability of nine reference genes by using four programs and confirmed that *18S rRNA* and *RPL7* were the most suitable single reference genes in spotted sea bass under normal and salinity stress conditions, respectively. Among different tissues, *ACTB* may be an appropriate choice when the target gene is not abundant in expression level. The best combination of reference genes was *RPL7* and *RNAPol II* according to overall ranking under salinity stress. Overall, the present study provides valuable information about the reference genes of *L. maculatus* that could be used for gene expression normalization in other teleost species.

## ACKNOWLEDGEMENTS

We are greatly appreciative of the laboratory members for assistance with feeding the fish and collecting the tissue samples.

### Funding

The present study was financially supported by the National Natural Science Foundation of China (NSFC, 31602147), Shandong Provincial Natural Science Foundation, China (ZR2016CQ21), the key laboratory of Mariculture (KLM), Ministry of Education, OUC (NO:2018008) and China Agriculture Research System (CARS-47). The funders had no role in study design, data collection and analysis, decision to publish, or preparation of the manuscript.

### Grant Disclosures

The following grant information was disclosed by the authors:
National Natural Science Foundation of China: NSFC, 31602147.
Shandong Provincial Natural Science Foundation: ZR2016CQ21.
The key laboratory of Mariculture (KLM), Ministry of Education, OUC: (NO:2018008).
China Agriculture Research System: CARS-47.

### Competing Interests

The authors declare there are no competing interests.

### Author Contributions

- Haolong Wang conceived and designed the experiments, performed the experiments, analyzed the data, contributed reagents/materials/analysis tools, prepared figures and/or tables, authored or reviewed drafts of the paper.
- Haishen Wen, Kaiqiang Zhang and Yang Liu contributed reagents/materials/analysis tools.
- Yun Li conceived and designed the experiments, authored or reviewed drafts of the paper, approved the final draft.

### Animal Ethics

The following information was supplied relating to ethical approvals (i.e., approving body and any reference numbers):

The Institutional Animal Care and Use Committee of the Ocean University of China and the China Government Principles.

### DNA Deposition

The following information was supplied regarding the deposition of DNA sequences:

The sequences are available in the Supplemental File and also at GenBank: BankIt2102897 Seq1 MH181798, BankIt2102934 Seq2 MH181799, BankIt2102938 Seq3 MH181800, BankIt2102940 Seq4 MH181801, BankIt2102942 Seq5 MH181802, BankIt2102943 Seq6 MH181803, BankIt2102945 Seq8 MH181804, BankIt2103576 Seq7 MH181805.

### Data Availability

The raw data are provided in the Supplemental Files.

### Supplemental Information

Supplemental information for this article can be found online at http://dx.doi.org/10.7717/peerj.5631#supplemental-information.

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
