# Peer review of "Evaluation of potential reference genes for quantitative RT-PCR analysis in spotted sea bass (Lateolabrax maculatus) under normal and salinity stress conditions"

_PeerJ, doi:10.7717/peerj.5631_

## Round 0.1 · original submission · Major Revisions

Dear Dr. H. Wang

After reading the comments of three independent reviewers, it is my opinion that the MS can still be improved before being published in Peer J. Despite the minor revisions, I would recommend that you pay particular attention on:

- Scientific name of the seabass species
- Clarification of the experimental design and salt stress conditions
- Description of qPCR according to the MIQE guidelines
- Clarification of the overall ranking criteria used by RefFinder
- Discussion of the results based on seabass papers, e.g. European seabass (Mitter et al., 2009) and Asian seabass (Paria et al., 2016)

I will be glad to receive a new version of the MS including all aspects referred by the reviewers,or otherwise a point-by-point justification for not including some of them,

Sincerely,

Ana I. Ribeiro-Barros

Reviewer 1 ·

Basic reporting

• This study by Wang et.al. on evaluation of suitable reference genes for qPCR analysis of gene expression in different tissues and under salinity stress is sufficiently comprehensive and important as far as qPCR studies on L. maculatus are concerned, which is an important commercial fish species from the coast of China. However, I would like to suggest few changes listed below in order to make the manuscript suitable for publication in PeerJ.

• The two major issues with this manuscript are with the title itself. Firstly, the scientific name for spotted sea bass is Dicentrarchus punctatus. However, authors mentioned spotted sea bass as Lateolabrax maculatus. Why? If it is Lateolabrax maculatus indeed, then it should be mentioned as Asian sea bass or Asian spotted sea bass, and not spotted sea bass (?). Please clarify.

• Secondly, Lateolabrax maculatus is an unaccepted species name according to World Register of marine species (WoRMS) database. The accepted name is suggested as Lateolabrax japonicus or Holocentrum maculatum. There have been some revisions of the exact species name after Kim et.al 2001 cited by authors (such as, Yokogawa 2013). As the study deals with single species, it is very important to have an accurate species name and identification. Please clarify how the species was identified as what it is claimed in the paper? Are both morphological and molecular criteria for species identification were verified and were congruent?


• In the abstract a sentence about why L. maculatus is missing. To give readers an idea upfront that it isn’t any random animal but a commercially important fish.

• Reporting stability values for NormFinder analysis is OK, however, for complete visualization of data, graphical representation of intra and intergroup difference is required so as to understand which pair of genes will balance the errors in expression estimates (for example see, G.G. Shimpi et al. / Journal of Experimental Marine Biology and Ecology 483 (2016) 42–52). This is a major advantage and an important component of NormFinder analysis (at least include this in the supplementary information).

• There are more relevant and recent examples of GAPDH being unstable. Citing tomato (plant) article in fish paper (marine animal) is quite odd.

Experimental design

Experimental design is robust and inclusion of several samples does provide biological significance. There are few issues that should be addressed, however.

Line 154 final extension at 72°C for 2 min. I don’t think qRT-PCR needs final extension. Also 2 min extension time is too long for PCR fragments > 200 bp.

It is not mentioned whether melting curve analysis was performed at the end of each qRT-PCR run in the “Primer design and qRT-PCR” section in methods. This is required to verify primer specificity in each run.

Whether oligo(dT) or random hexamers were used for cDNA synthesis is a very basic information and must be mentioned, which is missing.

Information on exact number of biological replicates is unclear from description in section 1 of Materials and Methods. The exact n=? is missing in the methods sections.

Validity of the findings

The analyses are robust and so are the number of replicates used. Hence the findings appear valid.

Additional comments

There are few places where grammatical mistakes and typos can be found. I would suggest few changes listed below.

In the title of the manuscript the species name for the spotted sea bass is written as “maculatus” whereas everywhere in the text it is written as “maculates”. Please write the correct species name without any spelling mistake in the entire text.

Line 51 for reference genes → as reference genes
Line 63 common used tool → commonly used tool
Line 83 to 86 Please rephrase the sentence. There is a word missing.
Line 97 endure challenges of salinity → endure changes in salinity
Line 104 to select and evaluate the stability in different tissues (incomplete sentence)
Line 113 tricaine methane sulfonate → 
tricaine methanesulfonate
Line 121 3 fish per group → 3 fishes per group 

Line 144 All reference genes were designed → All reference gene primers were designed
Line 151-152 First TaKaRa is mentioned as a supplier and later Tli RNAse Plus for the same SYBR Green, why? Please check.
Line 176 indicate what is CV? It appeared for the first time in the text.
Line 274-275. This sentence doesn’t make sense. Kindly revise.
Line 311 conclusion → Conclusion
Figure 3 What do the dotted lines indicate should be mentioned in the figure legend for reader’s understanding?

Kindly explain “Different salinities n=36” in Table 3. In the methods section it is mentioned that there are 4 groups with different salinities and 3 fishes per group were randomly selected, which means 4*3=12. So how this number “n=36” came about?

Reviewer 2 ·

Basic reporting

no comment

Experimental design

no comment

Validity of the findings

no comment

Additional comments

The paper was logically and English-fluently written, and was presenting the first study to select reference genes of L. Maculates by qRT-PCR. The results are significant and well presented, and this work should provide valuable information for future researches of selecting appropriate reference gene in L. maculates and other teleost species.

·

Basic reporting

Please find the attached review report.

Experimental design

Please find the attached review report.

Validity of the findings

Please find the attached review report.

Additional comments

Please find the attached review report.

---

## Round 0.2 · accepted · Accept

Dear Dr Wang,
Thank you for re-submitting the revised version of your MS. This has been revised by the two independent reviewers, which were very positive about the new version. Therefore, it is my pleasure to inform you that the paper is now accepted for publication in Peer J.

Sincerely
Ana Ribeiro-Barros

# Reviewer 1 ·

Basic reporting

The manuscript has definitely improved in the revised version with respect to basic reporting.

Experimental design

Experimental design is more clear now.

Validity of the findings

Validity of the findings is sound and robust in the revised version.

Additional comments

I am satisfied with the revision and I think the manuscript is ready for the publication.

·

Basic reporting

The manuscripts meets the requirement

Experimental design

The manuscripts meets the requirement

Validity of the findings

The manuscripts meets the requirement

Additional comments

Authors have addressed all the comments made by the reviewer.